# SplicedVAE: Learning Splicing Ratios from scRNA-seq to Enhance RNA Velocity and Cellular Trajectories

## 1 Introduction

Understanding cellular dynamics from static snapshots remains a fundamental challenge in single-cell biology. Traditional approaches such as scVelo (Bergen et al., 2020) attempt to infer RNA velocity, or the rate and direction of transcriptional change, by modeling the relationship between spliced and unspliced (S/U) mRNA counts (La Manno et al., 2018). Such models enable the prediction of future cell states and reconstruction of developmental trajectories. However, interest in inferring RNA velocity has surged faster than technology can keep pace. Most widely-used scRNA-seq protocols (particularly 3'-tagging methods) and datasets do not capture S/U information, severely limiting applicability. Recent computational approaches have attempted to predict velocity without S/U counts (Zeng et al., 2022; Mahajan & Maslov, 2024), but their performance remains limited by rigid assumptions of cellular kinetics, reliance on discrete rather than continuous predictions, and the inability to generalize across diverse biological contexts and modalities.

Beyond unreliable velocity prediction, existing methods are also unable to model cellular trajectories. Supervised approaches like Velo-Predictor (Wang & Zheng, 2021) and TFvelo (Li et al., 2024) predict discretized velocity directions but lack the generative structure needed to model the underlying stochastic dynamics of cell state transitions. Velocity-aware trajectory inference tools like CellRank (Lange et al., 2022) combine RNA velocity with manifold connectivity but are constrained to observed data and cannot extrapolate to unseen states. Meanwhile, generative models like diffusion-based methods (Luo et al., 2024; Ho et al., 2020) and flow-matching approaches (Klein et al., 2025) have shown promise for capturing complex scRNA-seq distributions but have primarily been applied to generate static cell profiles rather than model temporal dynamics.

In response to these limitations, we propose SplicedVAE, a supervised generative framework that learns splicing ratios directly from total gene expression counts through multitask variational autoencoders. By augmenting the scVI framework (Lopez et al., 2018) with an auxiliary decoder head for continuous splicing ratio prediction, our model leverages multitask learning to capture variation in cellular dynamics that expression-only models miss. This represents a key step toward unified generative frameworks capable of not only *reconstructing* but also *generating* novel cellular trajectories, enabling in silico perturbation experiments and prospective experimental design.

## 2 Methods

Our approach extends the scVI variational autoencoder by adding a secondary task: predicting the ratio of spliced to total mRNA for each gene. The intuition is straightforward—if we train a model to both reconstruct gene expression counts (the standard scVI objective) and predict splicing ratios (our auxiliary objective), the model's internal representation must encode information about cellular dynamics that expression alone cannot capture. By combining these objectives during training, the model learns a latent space that is better structured for downstream trajectory inference and velocity estimation. We evaluate this approach on pancreatic endocrinogenesis data, where ground-truth splicing information is available for validation, and compare against standard scVI and existing velocity prediction methods.

## 2.1 MODEL ARCHITECTURE

The full SplicedVAE architecture includes: (1) an scVI encoder capturing nonlinear gene–gene dependencies, (2) a negative binomial (NB) decoder reconstructing the gene-expression likelihood, and (3) a custom MLP decoder head (2-layer, 3-layer, or bottleneck architecture) to predict splicing ratios. The joint training objective combines the evidence lower bound (ELBO) for reconstruction with a weighted splicing prediction loss:

$$\mathcal{L} = \mathcal{L}_{\text{ELBO}} + \lambda\mathcal{L}_{\text{splice}}$$

where $\lambda$ controls the contribution of the auxiliary task. Multiple loss formulations were evaluated for $\mathcal{L}_{\text{splice}}$, including MSE, weighted MSE (gene-specific weighting by total counts), binomial likelihood, and L1 loss. This multitask formulation encourages the latent representation to encode information relevant to both gene expression and splicing dynamics, acting as an effective regularizer that improves reconstruction ELBO and trajectory recovery.

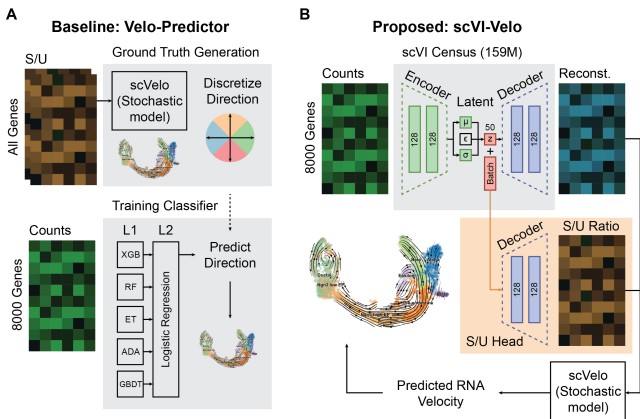

Figure 1: (A) The baseline Velo-Predictor pipeline. Genes with S/U counts are preprocessed with the stochastic model from scVelo and discretized into 4 classes as ground truth labels. An ensemble of models uses raw counts followed by logistic regression to predict discretized UMAP velocity directions. (B) Architecture of our proposed SplicedVAE method. Based on scVI, an S/U prediction head is added as a separate decoder for predicting splicing ratios. The stochastic scVelo model is applied to compute RNA velocity from predicted S/U ratios.

## 2.2 DATASETS AND PREPROCESSING

We use scRNA-seq data from two sources. For large-scale experiments, we leverage the Arc Virtual Cell Atlas scBaseCount resource (Youngblut et al., 2025), which provides raw gene-level counts together with Velocyto-derived S/U matrices across 230M cells from 21 species. For computational tractability, we restrict to *Homo sapiens*, 10x 3' or 5' protocols, datasets with >3,000 cells, and apply weighted round-robin sampling ($\alpha = 0.5$) across 78 tissues to yield a balanced corpus of ~10 million cells.

For controlled experiments and ablation studies, we use the pancreas 15-day endocrinogenesis dataset, a well-characterized developmental system with annotated cell types and known trajectories. We retain relevant covariates (sample, batch, species) to account for technical variation. When S/U layers are available, we compute stochastic scVelo velocities (Bergen et al., 2020) as estimates of transcriptional dynamics, serving as auxiliary supervision during training and ground truth for validation.

## 2.3 TRAINING CONFIGURATION

To minimize training costs, we initialize the encoder and NB decoder from pretrained scVI models trained on CellXGene Census data (release 2025-11-08). For the pancreas dataset, we use the *Mus musculus* model (43.7M cells); for Arc Atlas experiments, we use the *Homo sapiens* model (159M cells). The splicing decoder head is trained from scratch.

For the pancreas dataset, we adopt a standardized configuration: 150 epochs, 70:30 train/validation split, latent dimensionality of 50, encoder and decoder hidden layers of size 128. Unless otherwise noted, $\mathcal{L}_{\text{splice}}$ uses MSE with unit weight ($\lambda = 1$), and the S/U head uses a 2-layer MLP architecture. Hyperparameter optimization was performed via Optuna grid search over 90 configurations, varying architecture type (2-layer, 3-layer, bottleneck), activation (sigmoid, softmax), latent dimensionality (8–128), dropout (0–0.5), loss type, optimizer (Adam, AdamW, RMSprop), learning rate, and splicing loss weight. The top 5 models by test-set RMSE were retrained for 50 epochs.

## 2.4 Evaluation Metrics

We benchmark SplicedVAE on held-out cells from the pancreas dataset, quantitatively assessing: (1) splicing ratio prediction via RMSE and Pearson correlation between predicted and ground-truth ratios; (2) cosine similarity between velocity vectors derived from predicted versus true S/U counts; (3) directional classification accuracy for a simplified 4-class velocity task, comparing against Velo-Predictor (Wang & Zheng, 2021) and standard scVI without the splicing head. Qualitative evaluation involves inspecting UMAP embeddings and RNA velocity streamlines to assess whether known developmental trajectories of pancreatic endocrine lineages are faithfully recovered.

## 3 Results

For splicing ratio prediction, our model achieved a test-set RMSE of 0.1271 and a positive correlation (R=0.67) between predicted and ground-truth splicing ratios on held-out test cells. This indicates the capture of meaningful signal for our *auxiliary* task. Further analysis of our primary task of creating a better latent space for cellular dynamics reveals meaningful results.

## 3.1 Improved Latent Representations via Multitask Learning

To isolate the contribution of the splicing prediction task, we trained a standard scVI model without the auxiliary head on the same pancreas dataset. Figure 2 (Appendix) reveals that while scVI successfully clusters major cell types in UMAP space, trajectory reconstruction from the latent space alone fails to capture coherent developmental flows (Fig. 2C-D). When trained on a gene subset overlapping with scVelo's features (5706/8000 genes), reconstruction quality degrades further, with Alpha, Beta, and Delta clusters exhibiting disorganized trajectories (marked in red circles; Fig 2D). Reconstruction ELBO loss plateaus at higher values (Fig. 2E), and the inferred differentiation path deviates from known biology (Fig. 2F).

In contrast, Figure 3 (Appendix) reveals SplicedVAE achieves lower reconstruction ELBO loss (Fig. 3B), demonstrating that the splicing prediction head acts as an effective regularizer. The learned latent space exhibits clearer cluster boundaries and better cell type separation compared to standard scVI (Fig. 3G vs. Fig. 2B). High Pearson correlation (Fig. 3D-E) and cosine similarity (Fig. 3F) between predicted and ground-truth velocity vectors indicate that SplicedVAE captures meaningful directional information exceeding random baselines.

## 3.2 Velocity Field Reconstruction

Velocity streamlines derived from SplicedVAE's predicted splicing ratios (Fig. 3I) largely recapitulate the directional patterns observed in ground-truth velocity fields computed from true S/U counts (Fig. 3H). The model preserves key trajectory structures across cell type clusters in UMAP space, suggesting successful capture of underlying developmental dynamics. However, localized discrepancies remain: several regions show disorganized flow patterns (red circles in Fig. 3H), and inferred pseudotime occasionally contradicts known biological progression, indicating room for improvement.

Gene-level velocity comparisons (Fig. 3L for gene Cpe) and high-level trajectory maps (Fig. 3M) further illustrate the model's ability to reconstruct biologically plausible differentiation flows. Directional classification accuracy reaches 50% (Fig. 3K), which falls short of the 88% baseline performance from Velo-Predictor (Wang & Zheng, 2021). While this suggests that continuous splicing ratio prediction does not yet outperform discrete directional classifiers, the continuous predictions

enable downstream velocity estimation via scVelo, providing a more flexible representation than categorical outputs.

# 4 DISCUSSION AND FUTURE WORK

We introduce SplicedVAE, a novel supervised generative framework that learns splicing dynamics directly from total gene expression counts through multitask variational autoencoders. To our knowledge, this is the first approach to successfully predict continuous splicing ratios from expression-only data while simultaneously improving latent representations for trajectory inference. The key innovation is leveraging splicing prediction as an auxiliary task that regularizes the latent space to encode dynamics-relevant variation that expression-only models cannot capture, leading to more structured representations and more coherent velocity fields.

Our results demonstrate three core contributions: (1) moderate but significant correlation (R=0.67) between predicted and ground-truth splicing ratios without requiring S/U sequencing, (2) improved reconstruction ELBO and clearer cell type clustering through multitask learning, and (3) velocity fields that recapitulate known developmental trajectories in pancreatic endocrinogenesis. These findings suggest that splicing information—though noisy—provides a complementary signal that enhances both generative modeling and dynamical inference in single-cell data.

Our immediate priorities focus on scaling and refinement. First, we will extend experiments to the full Arc Virtual Cell Atlas (∼10M cells across 78 tissues), evaluating whether patterns learned in pancreas generalize to diverse developmental and homeostatic systems. This will test whether a single pretrained model can predict splicing ratios across tissues and species, or whether tissue-specific fine-tuning is necessary.

Future work aims to implement diffusion-based and flow-matching architectures to move beyond point predictions of splicing ratios. Instead of predicting a single velocity vector per cell, diffusion models can learn distributions over possible future states, capturing uncertainty in cell fate decisions. We also plan to incorporate multimodal data (e.g chromatin accessibility or methylation datasets) to constrain and improve splicing predictions.

## MEANINGFULNESS STATEMENT

Cellular dynamics govern development, differentiation, and disease progression. Meaningful biological representations must capture not only static cellular states but also the temporal transitions between them. SplicedVAE addresses this directly by learning representations that encode splicing dynamics—a direct readout of transcriptional regulation—from widely available expression data. By enabling velocity estimation in previously incompatible datasets spanning millions of cells, this work expands the scope of single-cell modeling, bringing us closer to predictive frameworks and foundation models for cellular behavior that transforms *descriptive* biology into *predictive* biology, where in silico models guide hypothesis generation and experimental design at scale.

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

## A  APPENDIX: FIGURES & VISUALIZATIONS

### A.1  SPLICEDVAE IMPROVES LATENT REPRESENTATIONS

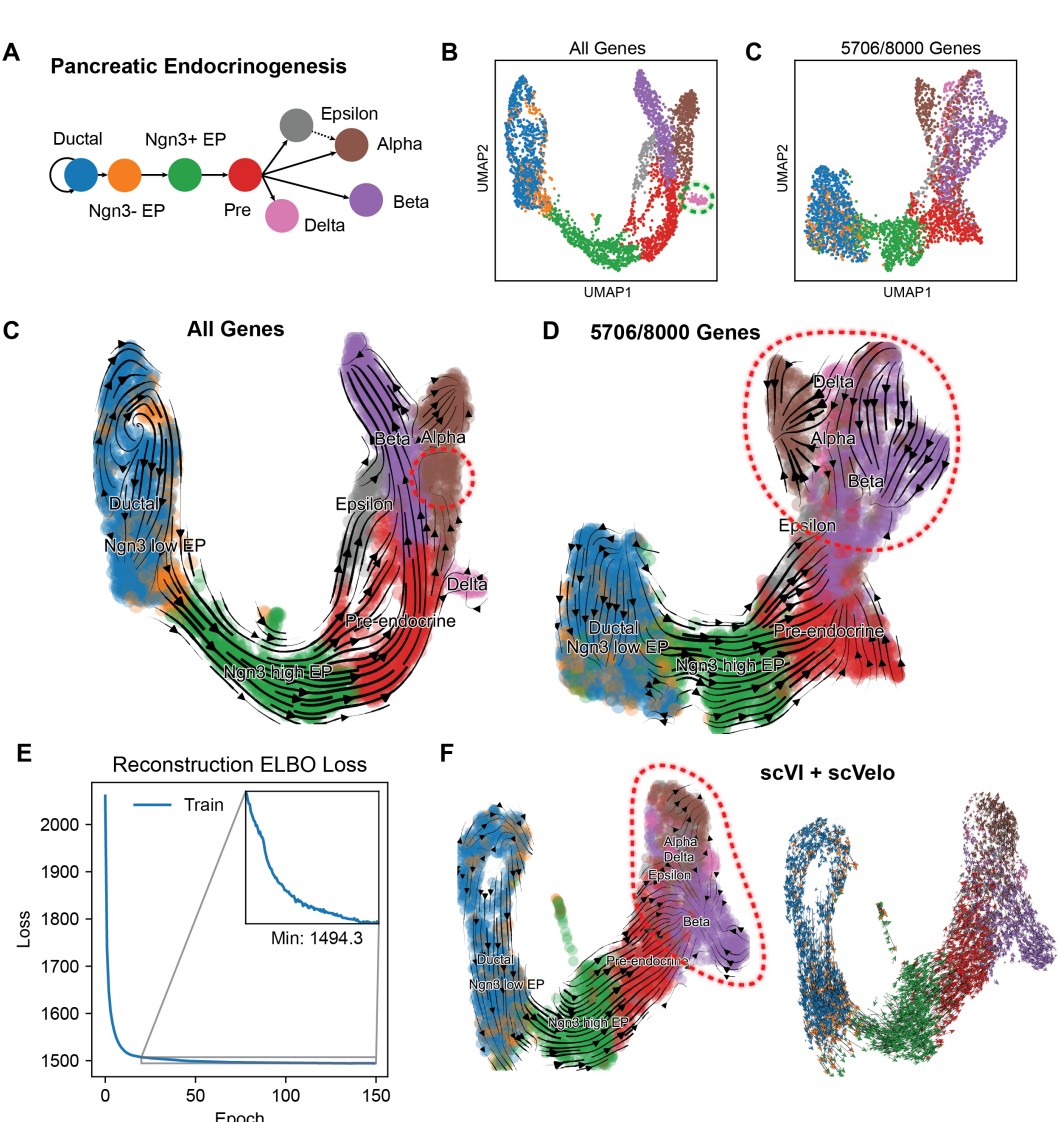

Figure 2: Baseline scVI model limitations. (A) Known developmental hierarchy of pancreatic endocrine cells. (B) UMAP embedding of scVI latent space for full and subset gene sets. (C-D) Trajectory reconstructions for all genes and gene subset, showing disorganized flow patterns (red circles) in Alpha, Beta, and Delta clusters when trained on fewer genes. (E) Reconstruction ELBO loss during scVI training, failing to minimize sufficiently. (F) UMAP embedding from standard scVI, demonstrating inaccurate trajectory inference without velocity information.

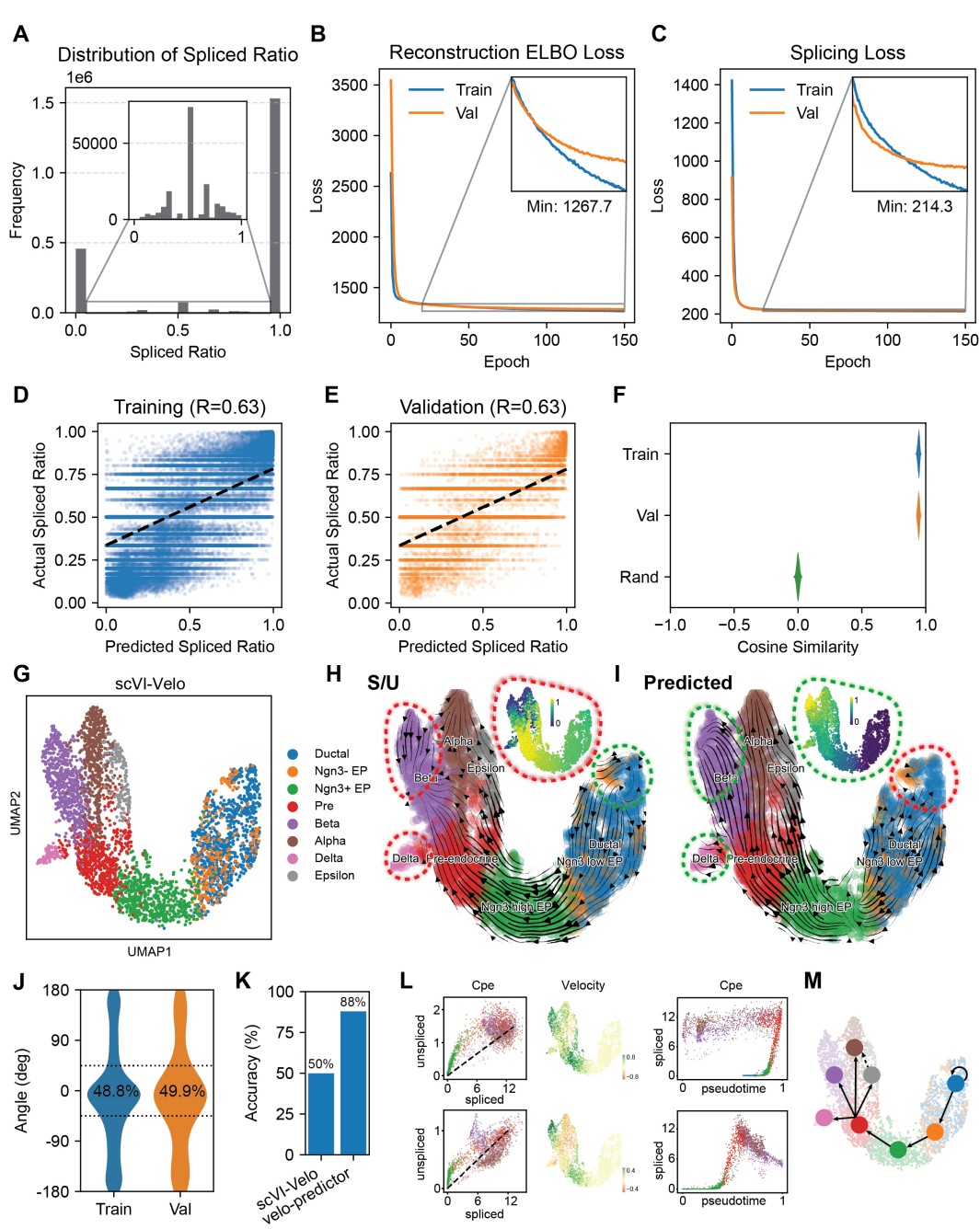

Figure 3: SplicedVAE training dynamics and performance. (A) Distribution of splicing ratios across genes and cell types. (B) Reconstruction ELBO loss and (C) splicing ratio prediction loss during training and validation. Pearson correlation between predicted and ground-truth splicing ratios during (D) training and (E) validation. (F) Cosine similarity between predicted and true velocity vectors, compared to random baseline. (G) UMAP embedding of SplicedVAE latent space, colored by cell type. Velocity streamlines computed from (H) ground-truth S/U counts and (I) SplicedVAE predictions. (J) Distribution of velocity angles in UMAP space. (K) Directional classification accuracy for SplicedVAE vs. Velo-Predictor. (L) Gene-level velocity comparison (Cpe). (M) High-level differentiation trajectory map.

