# OpenReview forum: "SplicedVAE: Learning Splicing Ratios from scRNA-seq to Enhance RNA Velocity and Cellular Trajectories"
_ICLR.cc/2026/Workshop/LMRL — Submitted to ICLR 2026 Workshop LMRL_

### Official Review · Reviewer_pp3w · 2026-02-22
**Review of Submission 24**

**Rating:** 6
**Confidence:** 3

**Review:**

Summary
This tiny paper submission proposes SplicedVAE, a method that involves fine-tuning scVI on mRNA splicing ratios (the ratio of spliced counts total counts). This fine-tuning is intended to help improve latent representations' ability to model time-dependent trajectories, while enabling it to predict RNA velocity from just static counts.
Strengths
- The proposed method is a sensible solution to a well-motivated problem. Splicing prediction as an additional signal to encourage a dynamics-informed refinement of the latent space is biologically sensible; motivation is consistent with other multi-modal representation work
- The framework allows simultaneous representation learning and prediction of splicing ratios with a single model (as opposed to training separate models on the two tasks).

Weaknesses
- The most significant weakness of this work is its unconvincing experimental results. The paper leans quite heavily on qualitative analyses, where the authors show improvements via UMAP visualizations and velocity streamlines in the UMAP space. However, UMAP is quite sensitive to hyperparameters and stochasticity, which potentially casts some doubt on some of the results. The few quantitative results reported are also unconvincing: the directional classification accuracy is signficantly worse than the baseline (0.5 vs 0.88), and comparing the ELBO on the training set is not particularly meaningful.
- Could benefit from some additional literature review on other biological multi-modal representation learning efforts, which are conceptually closely related, albeit not with these specific data modalities.

Recommendation
- Despite the unconvicing empirical results, the proposed method is sensible. This tiny paper provides a reasonable proof of concept, and with some additional development and validation, has the potential to be a strong work. Because of this, I am rating it a 6.

---

### Official Review · Reviewer_L9Gc · 2026-02-23
**Continuous splicing ratios prediction with SplicedVAE**

**Rating:** 5
**Confidence:** 3

**Review:**

The authors of submission24 present SplicedVAE to infer splicing ratios from scRNA-seq count data. The model is based on the scVI framework with an additional head to predict the spliced and unspliced (S/U) ratios and further compute RNA velocity. The presentation is clear and the flow is relatively easy to follow. There are a few things I would like to point out:
- While the authors claim that this is the first approach to predict continuous S/U ratios, their accuracy of 50% is significantly lower than 88% from the Velo-Predictor baseline. The authors also did not provide an explanation for the potential reasons of this gap. Does continuous/discrete predictions affect the calculation of directional classification, which unwillingly places their method in a disadvantageous place when compared to the baseline method?
- The authors include detailed qualitative results and visualizations in Figures 2 and 3 as well as some numbers in the main text. However, a table with numbers will present their performance more clearly. Although the authors calculated 4 evaluation metrics to evaluate the model, it's now difficult to track how well SplicedVAE performs against baseline models quantitatively. Also, it looks like that all 4 metrics are not calculated for all baseline methods.
- The future works look promising with diffusion and flow matching methods. However, will these improve the directional accuracy? Or are there any plans to investigate and resolve the accuracy gap? If the gap still exists, it may be challenging to convince users to move to continuous predictors from existing discrete predictors.
- Data preparation details are missing from the paper. I inferred that they followed the scVI workflow.
- Is scVI-Velo an alternative name of SplicedVAE? Those two names are being used interchangeably in the paper. For example, Figures 1B, 2F, and 3K.

---

### Official Review · Reviewer_EAT9 · 2026-02-25
**Needs major revision and resubmission**

**Rating:** 4
**Confidence:** 5

**Review:**

SplicedVAE extends scVI by adding an auxiliary decoder head that predicts spliced-to-total mRNA ratios from total gene expression counts. The model is trained with a multitask objective: standard ELBO reconstruction plus a weighted splicing prediction loss.

Strengths:
The idea that splicing prediction acts as a regularizer for latent space structure is reasonable and empirically supported by lower ELBO and clearer UMAP separation. Unlike Velo-Predictor’s discrete velocity classes, this approach outputs continuous splicing ratios, allowing downstream flexible velocity modeling.

Weknesses:
- The overall learning steps are: Expression -> predict splicing ratio -> feed to scVelo -> compute velocity. The model does not directly learn velocity,  temporal transitions and does not model transcriptional dynamics. It only predicts a ratio proxy and relies entirely on scVelo’s kinetic model afterward. This limits methodological novelty.

- The claim is that multitask learning improves latent structure. But there is no rigorous quantitative demonstration that latent space encodes true temporal structure.

- The model performs substantially worse on a core velocity metric. The argument that continuous outputs are more flexible does not compensate for lower accuracy. This is a serious methodological weakness.

- RNA velocity is inherently stochastic. The model outputs point estimates, provides no predictive uncertainty. This limits interpretability and biological usefulness.

- Hyphenation is inconsistent (e.g., “cell type” vs. “cell-type,” “gene expression” vs. “gene-expression”). Please standardize.
- The experimental evaluation relies heavily on qualitative UMAP and streamline visualizations; more rigorous quantitative trajectory metrics would strengthen the results.
- Results are primarily shown on a single pancreas dataset; cross-dataset validation would improve generalizability claims.

---

### Meta-Review · Area_Chair_3xGi · 2026-02-25

**Recommendation:** Reject
**Confidence:** 3

**Metareview:**

I recommend rejection.

---

### Decision · Program_Chairs · 2026-03-02

**Decision:**

Reject

**Comment:**

Please see the meta-review.